# Probabilistic, entropy-maximizing control of large-scale neural synchronization

**Melisa Menceloglu[1]¤, Marcia Grabowecky[1,2], Satoru Suzuki [1,2]***

**1** Department of Psychology, Northwestern University, Evanston, IL, United States of America,
**2** Interdepartmental Neuroscience, Northwestern University, Evanston, IL, United States of America

¤ Current address: Cognitive, Linguistic, and Psychological Sciences, Brown University, Providence, RI, United States of America
* satoru@northwestern.edu

**Data Availability Statement:** The relevant EEG data are available at: https://doi.org/10.21985/n2-8hxs-md53.

**Funding:** National Institutes of Health T32 NS047987 to MM.

## Abstract

Oscillatory neural activity is dynamically controlled to coordinate perceptual, attentional and cognitive processes. On the macroscopic scale, this control is reflected in the U-shaped deviations of EEG spectral-power dynamics from stochastic dynamics, characterized by disproportionately elevated occurrences of the lowest and highest ranges of power. To understand the mechanisms that generate these low- and high-power states, we fit a simple mathematical model of synchronization of oscillatory activity to human EEG data. The results consistently indicated that the majority (~95%) of synchronization dynamics is controlled by slowly adjusting the probability of synchronization while maintaining maximum entropy within the timescale of a few seconds. This strategy appears to be universal as the results generalized across oscillation frequencies, EEG current sources, and participants ($N$ = 52) whether they rested with their eyes closed, rested with their eyes open in a darkened room, or viewed a silent nature video. Given that precisely coordinated behavior requires tightly controlled oscillatory dynamics, the current results suggest that the large-scale spatial synchronization of oscillatory activity is controlled by the relatively slow, entropy-maximizing adjustments of synchronization probability (demonstrated here) in combination with temporally precise phase adjustments (e.g., phase resetting generated by sensorimotor interactions). Interestingly, we observed a modest but consistent spatial pattern of deviations from the maximum-entropy rule, potentially suggesting that the mid-central-posterior region serves as an "entropy dump" to facilitate the temporally precise control of spectral-power dynamics in the surrounding regions.

## Introduction

A great deal of evidence suggests that the coordination of oscillatory activity contributes to controlling neural communications that are necessary for effective operations of perception, attention, memory, and cognition [e.g., 1–18]. While fine-tuned coordination likely involves controlling the phases of oscillatory activity across frequency bands, the impact of oscillatory activity also depends on the size of synchronized neural population. For large-scale neural activity detected by scalp-recorded electroencephalography (EEG), the spectral power

**Competing interests:** The authors have declared that no competing interests exist.

obtained at an EEG current source reflects the size of synchronously oscillating population within its spatial resolution. EEG spectral power fluctuates at each current source reflecting the dynamic changes in large-scale synchronization of oscillatory activity. The goal of the current study was to elucidate the mechanisms that control these large-scale synchronization dynamics.

One way to investigate dynamic control is to compare spectral-power dynamics between EEG data and their phase-scrambled controls. Phase scrambling randomizes cross-frequency phase relations, thus destroying temporal structures that depend on cross-frequency phase alignment, rendering spectral-power dynamics stochastic (memory free) while preserving time-averaged power spectra. Because stochastic dynamics reflect a Poisson process, phase-scrambled spectral-power dynamics are characterized by an exponential temporal distribution of power. Actual EEG spectral-power dynamics deviate from exponential profiles in a characteristic U-shaped manner with disproportionately elevated occurrences of the lowest and highest ranges of power (see Results). This indicates that brain spectral-power dynamics exhibit intermittent bursts of extensive oscillatory synchronization separated by periods of sparse synchronization (compared with stochastic dynamics). How are these periods of extensive and sparse synchronization generated?

On the one hand, the brain neural network may actively boost large-scale synchronization or inhibit it in precise temporal coordination with behavioral demands. On the other hand, the network may indirectly influence large-scale synchronization by increasing or decreasing the probability of synchronization on a relatively slow timescale while generally maintaining maximum entropy for energy efficiency. As described in the results section, these possibilities can be evaluated in a relatively simple manner.

## Materials and methods

### Participants

Fifty-two Northwestern University students (35 women, 1 non-binary; ages 18 to 29 years, $M = 20.75$, $SD = 2.52$) gave informed written consent to participate for monetary compensation ($10/hr). All were right-handed, had normal hearing and normal or corrected-to-normal vision, and had no history of concussion. They were tested individually in a dimly lit or dark room. The study protocol was approved by the Northwestern University Institutional Review Board. Participants p1-p7 and p12-p28 ($N = 24$) participated in the rest-with-the-eyes-closed condition where their EEG was recorded for ~5 min while they rested with their eyes closed and freely engaged in spontaneous thoughts. This condition was always run first for those who also participated in the nature-video condition. Participants p8-p28 ($N = 21$) also participated in the nature-video condition where their EEG was recorded for ~5 min while they viewed a silent nature video. To evaluate the test-retest reliability, the nature-video condition was run twice (20–30 min apart), labeled as earlier viewing and later viewing in the analyses. A generic nature video was presented on a 13-inch, 2017 MacBook Pro, 2880(H)-by-1800(V)-pixel-resolution screen with normal brightness and contrast settings, placed 100 cm away from participants, subtending 16˚(H)-by-10˚(V) of visual angle. Participants p29-p52 ($N = 24$) participated in the replication of the rest-with-the-eyes-closed condition and the rest-with-the-eyes-open-in-dark condition which was the same as the former except that the room was darkened and participants kept their eyes open while blinking naturally.

### EEG recording and pre-processing

While participants rested with their eyes closed, rested with their eyes open in dark, or viewed a silent nature video for approximately 5 min, EEG was recorded from 64 scalp electrodes

(although we used a 64-electrode montage, we excluded signals from noise-prone electrodes, *Fpz*, *Iz*, *T9*, and *T10*, from analyses) at a sampling rate of 512 Hz using a BioSemi ActiveTwo system (see www.biosemi.com for details). Electrooculographic (EOG) activity was monitored using four face electrodes, one placed lateral to each eye and one placed beneath each eye. Two additional electrodes were placed on the left and right mastoid area. The EEG data were pre-processed using EEGLAB and ERPLAB toolboxes for MATLAB [19, 20]. The data were re-referenced offline to the average of the two mastoid electrodes, bandpass-filtered at 0.01 Hz-80 Hz, and notch-filtered at 60 Hz (to remove power-line noise that affected the EEG signals from some participants). For the EEG signals recorded while participants rested with the eyes open in dark or while they viewed a silent nature video, an Independent Component Analysis (ICA) was conducted using EEGLABs' *runica* function [21, 22]; blink related components were visually identified (apparent based on characteristic topography) and removed (no more than two components were removed per participant). We surface-Laplacian transformed all EEG data for the following reasons. The transform substantially reduces volume conduction (to within adjacent sites for a 64-channel montage [23]), virtually eliminates the effects of reference electrode choices (as we verified), and provides a data-driven method to fairly accurately map scalp-recorded EEG to current-source activities in the cortex [24–26]. We used Perrin and colleagues' algorithm [27–29] with the "smoothness" value, $\lambda = 10^{-5}$ (recommended for 64 channels [23]). We refer to the surface-Laplacian transformed EEG signals that represent the current sources under the 60 scalp sites (with the 4 noise-prone sites removed from analyses; see above) simply as EEG signals.

## EEG analysis

**EEG temporal derivative.**   An example 1 sec EEG waveform at a central site *FCz* from one participant (at rest with the eyes closed) is shown in Fig 1A (black curve). The mean spectral-amplitude profile of the full length (~5 min) version of the same data, with the fast Fourier transform (FFT) computed on each consecutive 5 sec waveform and then averaged, is shown in Fig 1B (black curve; the shaded area represents ±1 standard error of the mean). The general linear decrease in the spectral amplitude for higher frequencies with a slope of approximately 1 (in log-log scale) reflects the $1/f^{\beta}$ (with $\beta \sim 1$) background spectral profile largely explained by the neuronal Ornstein-Uhlenbeck process that exhibits a random-walk type behavior ([e.g., 30, 31]; random walk [integer or Gaussian] would yield $\beta = 1$). The spectral "bumps" seen around 10 Hz, 20 Hz, and 30 Hz indicate the characteristic bands of oscillation frequencies that the neural population reflected at this site for this person may utilize for communication and/or information processing. Taking the temporal derivative of EEG ($\frac{\Delta EEG}{\Delta t}$, where $\Delta t$ is the temporal resolution, i.e., 1/512 sec) (see the black curve in Fig 1C) highlights the oscillatory bumps by largely removing the $1/f^{\beta}$ background when $\beta \sim 1$ due to the trigonometric property of implicitly multiplying each frequency component by $f$ (see the black curve in Fig 1D). While Fig 1D shows an example at one site from one participant, we confirmed that taking the temporal derivative generally flattened the background spectral-amplitude profiles across sites and participants, indicating that our EEG data generally yielded $\beta \sim 1$ for their time-averaged spectral backgrounds. However, $\beta$ is known to fluctuate over time (see [32] for a review of the various factors that influence $\beta$; see [33] for contributions of the excitatory and inhibitory dynamics to $\beta$); thus, the degree to which taking the temporal derivative continuously reduces the $1/f^{\beta}$ background to highlight oscillatory activity is unclear. Nevertheless, we used the EEG temporal derivative, EEGd (as in our prior study [34]), because (1) EEGd is a "deeper" neural measure than EEG in the sense that scalp-recorded EEG potentials are caused by the underlying neural currents and taking the EEG temporal derivative macroscopically estimates those

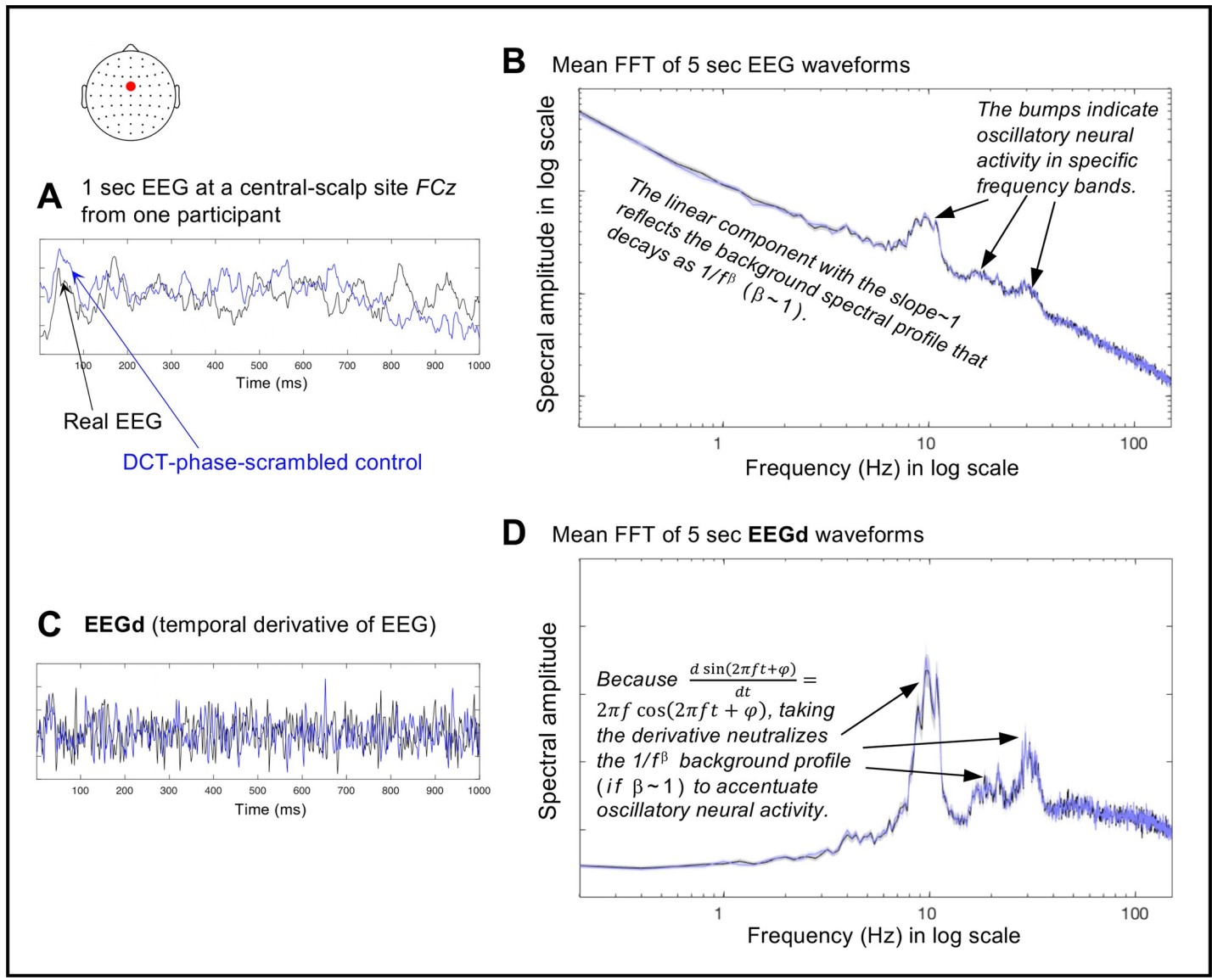

**Fig 1. The use of the temporal derivative of EEG (EEGd) and DCT-phase-scrambled controls for investigating the dynamic control of synchronization of oscillatory activity. A.** An example of 1 sec EEG waveform (black) and its DCT-phase-scrambled control (blue) at *FCz* from one participant. **B.** The mean spectral-amplitude profiles of the full length (~5 min) versions of the same EEG data (black) and its DCT-phase-scrambled control (blue), with the fast Fourier transform (FFT) computed on each consecutive 5 sec waveform and then averaged, plotted in a log-log format. **C.** The temporal derivatives, which we call EEGd, of the example EEG waveform (black) and its DCT-phase-scrambled control (blue) shown in A. **D.** The mean spectral-amplitude profiles of the full length (~5 min) versions of the same EEGd data (black) and its DCT-phase-scrambled control (blue), with the fast Fourier transform (FFT) computed on each consecutive 5 sec waveform and then averaged, plotted in a semi-log format. For B and D, the shaded areas represent ±1 standard error of the mean based on the FFTs computed on multiple 5 sec waveforms. The units are arbitrary (a.u.).

currents, (2) EEGd is virtually drift free (e.g., Fig 1C), and (3) it highlights the dynamics of oscillatory activity at least for temporal average (Fig 1D).

**Computing spectral power as a function of time.** The spectral-amplitude profiles shown in Fig 1B and 1D are time-averaged (standard fast Fourier transforms). To investigate how spectral power (amplitude squared of sinusoidal components) fluctuated, we used a Morlet wavelet-convolution method suitable for time-frequency decomposition of signals containing multiple oscillatory sources of different frequencies (see [23] for a review of different methods

for time-frequency decomposition). Each Morlet wavelet is a Gaussian-windowed sinusoidal templet characterized by its center frequency as well as its temporal and spectral widths that limit its temporal and spectral resolution. We decomposed each EEGd waveform into a time series of spectral power using Morlet wavelets with twenty center frequencies $f_c$'s between 6 Hz and 50 Hz, encompassing the $\theta$, $\alpha$, $\beta$, and $\gamma$ bands. The $f_c$'s were logarithmically spaced as neural temporal-frequency tunings tend to be approximately logarithmically scaled [e.g., 35, 36]. The accompanying $n$ factors (roughly the number of cycles per wavelet, $n = 2\pi f \cdot SD$, where $SD$ is the wavelet standard deviation) were also logarithmically spaced between 4.4 and 14.5, yielding the temporal resolutions ranging from $SD = 117$ ms (at 6 Hz) to $SD = 46$ ms (at 50 Hz) and spectral resolutions ranging from $FWHM$ (full width at half maximum) = 3.2 Hz (at 6 Hz) to $FWHM = 8.2$ Hz (at 50 Hz). These values strike a good balance for the temporal/spectral-resolution trade-off, and are typically used in the literature [e.g., 23].

**Generating phase-scrambled controls.** We generated phase-scrambled control data whose spectral power fluctuated stochastically (i.e., unpredictably in a memory free manner) while maintaining the time-averaged spectral-amplitude profiles of the actual EEG data. While phase-scrambling can be performed using several different methods, we chose discrete cosine transform, DCT [e.g., 37]. In short, we transformed each ~5 min EEG waveform with type-2 DCT, randomly shuffled the signs of the coefficients, and then inverse-transformed it with type-3 DCT (the "inverse DCT"), which yielded a phase-scrambled version. DCT phase-scrambling is similar to DFT (discrete Fourier transform) phase-scrambling except that it is less susceptible to edge effects. We verified that DCT phase-scrambling yielded a desired outcome, generating waveforms whose spectral-power fluctuations conformed to exponential distributions (see Results) indicative of a Poisson point process (a stochastic process), with virtually no distortions to the time-averaged spectral-amplitude profiles of EEG or EEGd (e.g., the blue curves overlap the black curves in Fig 1B and 1D).

**Computing entropy per interval, d.** We computed entropy for non-overlapping intervals of duration $d$ (sec). For each interval, we divided spectral power values into $N_{bins}$ bins using the Freedman-Diaconis method [38],

$$N_{bins} = ceil\left(\frac{max}{2 \cdot iqr \cdot [d \cdot 512]^{-1/3}}\right),$$

where $max$ is the maximum spectral-power value corresponding to the highest bin, $iqr$ is the interquartile range, $d \bullet 512$ is the number of spectral-power values available within each $d$ (sec) interval (sampled at 512 Hz), and $ceil$ takes the nearest larger integer. The value of $max$ was chosen such that the highest bin reached the 99.9th percentile or higher spectral-power value for each frequency and behavioral condition (because spectral-power values varied primarily as a function of frequency and condition). The $iqr$ values were computed per frequency per condition, averaged across frequencies, then averaged within the same behavioral condition, yielding three values, one for the rest-with-the-eyes-closed condition (averaged across the original and replication conditions), one for the rest-with-the-eyes-open-in-dark condition, and one for the nature-video condition (averaged for the earlier and later viewing conditions). Thus, $N_{bins}$ was optimized for the condition-specific $iqr$ and the number of data points within $d$ (sec) interval while the same range $[0, Max]$ was used in all cases. Using these spectral-power bins, we generated the probability distribution of spectral power values for each $d$ (sec) interval (per frequency per site per participant per condition), and computed the corresponding entropy using the standard equation,

$$S = -\sum_{i=1}^{i=N_{bins}} p_i \cdot ln(p_i),$$

where $S$ is entropy, $p_i$ is the proportion of spectral-power values within the $i$th bin, and $N_{bins}$ is the number of bins.

## Results and discussion

We started with a simple stochastic model of neural synchronization. As most neural connections are short-range [e.g., 39], we postulated that, at each moment, synchronization would sequentially spread from neural-unit to neural-unit with the probability $p_{term}$ that the rapid spreading would terminate at any given unit. It is reasonable to assume that the number of neural units is large and $p_{term}$ is small. Thus, the probability that the size of synchronized neural population, $N$ units, is larger than $n$ units at a given timepoint, is provided by the Poisson equation,

$$P(N > n) = \frac{(p_{term} \cdot n)^0}{0!} e^{-p_{term} \cdot n} = e^{-p_{term} \cdot n}. \qquad \text{Eq1}$$

Then, the probability density function $f(n)$ for the occurrence of a synchronized population of size $n$ can be obtained by solving,

$$\int_n^\infty f(n)dn = e^{-p_{term} \cdot n}, \qquad \text{Eq2}$$

yielding,

$$f(n) = p_{term} e^{-p_{term} \cdot n}. \qquad \text{Eq3}$$

It can be shown (e.g., using Lagrange multipliers) that, for any random variable ($n$ here), entropy (defined by the standard equation) is maximized for a given distribution average ($<n>$ here) when the probability density function is exponential. Therefore, the exponential form of $f(n)$ in *Eq* 3 indicates that the model yields maximum entropy for fluctuations in $n$ for a given temporal average $<n>$. Thus, our model (*Eq* 1) describes a simple macroscopic mechanism that generates synchronization dynamics that maximize entropy for a given temporal average (*Eq* 3). We note that any reasonable model that leads to an exponential probability density function for $n$ in conjunction with a parameter related to the probability of synchronization would be just as appropriate for our discussion.

It is reasonable to assume that EEG spectral power at a given site is proportional to the size of the synchronously oscillating neural population $n$ within the accessible current sources. Then, *Eq* 3 predicts an exponential distribution for the fluctuations of spectral power for phase-scrambled EEG (which are rendered stochastic). Our data confirmed this prediction (the thinner horizontal lines in Fig 2). Our goal was to elucidate the mechanisms that make the actual EEG spectral-power dynamics deviate from stochastic (exponential) dynamics in the characteristic U-shaped manner (the thicker curves in Fig 2). To this end, we considered the relationship between average spectral power and entropy.

Stochastic dynamics such as the spectral-power dynamics of the phase-scrambled controls (the thinner horizontal lines in Fig 2) are well fit by *Eq* 3 with a constant $p_{term}$. Nevertheless, the effective value of $p_{term}$ within an interval of duration $d$, which we call $p_{term.d}$, stochastically fluctuates with the variance given by,

$$Var(p_{term.d}) \propto p_{term} \cdot (1 - p_{term})/d. \qquad \text{Eq4}$$

Note that this is analogous to the familiar coin-tossing example. While the probability of getting heads is stationary with $p_{heads} = \frac{1}{2}$ for each (fair) coin toss, the effective probability of

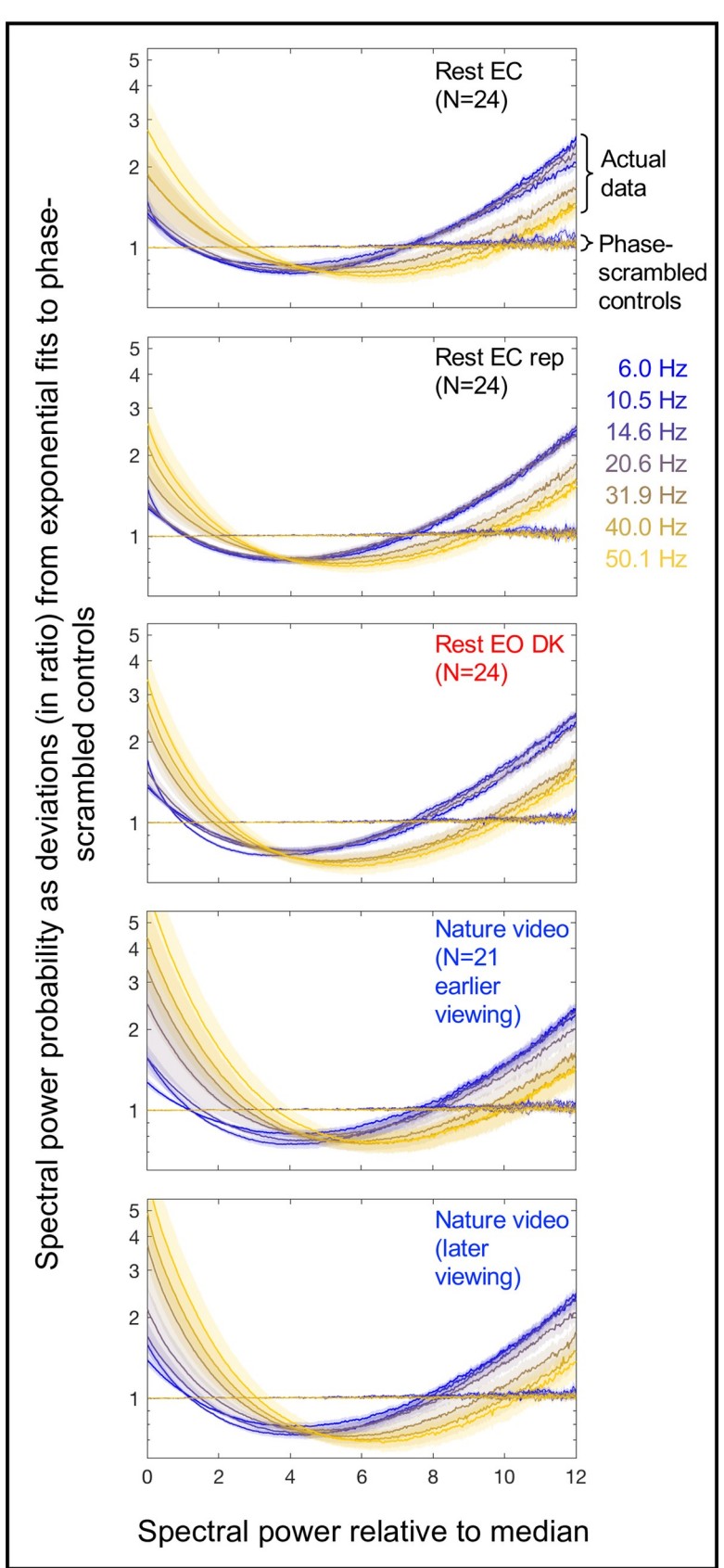

**Fig 2. Probability distributions of EEG spectral power as deviations (in ratio) from exponential fits to phase-scrambled controls.** Spectral-power probability distributions (using 500 bins) are shown for the actual EEG data and their phase-scrambled controls for representative frequency bands, $\theta$ (6.0 Hz), $\alpha$ (10.5 Hz), $\beta$ (14.6 Hz and 20.6 Hz) and $\gamma$ (31.9 Hz, 40.0 Hz, and 50.1 Hz), color-coded from cooler to warmer. On the x-axis, spectral power is normalized to the median power per frequency per site per condition per participant. The probability distributions are averaged across sites and participants. The y-axis indicates spectral-power probability relative to the exponential fits to the corresponding phase-scrambled controls. For example, the value 1 indicates that the probability of a specific spectral-power value was as predicted by the exponential fit, 2 indicates that the probability was twice predicted by the exponential fit, etc. Note that all probability distributions for the phase-scrambled controls (thinner lines) were exponential, tightly conforming to the line of y = 1. The five panels show the probability distributions for the five conditions: ~5-min rest with the eyes closed (Rest EC), its replication (Rest EC rep), ~5-min rest with the eyes open in dark (Rest EO DK), and the earlier and later ~5-min viewing of a silent nature video (Nature video). The distributions for the actual EEG data (thicker lines) deviate from the exponential form in a characteristic U-shaped manner with elevated occurrences of the lowest and highest ranges of power. The shaded areas represent ±1 standard error of the mean with participants as the random effect.

getting heads, that is, the actual proportion of heads obtained for a given set of $N$ tosses, $p_{heads,N}$, is variable, with its set-to-set variance given by, $Var(p_{heads,N}) = p_{heads}\cdot(1-p_{heads})/N$.

For stochastic dynamics of synchronization, *Eq 3* holds within an interval of any duration (given that it includes sufficient data points to reliability evaluate the probability distribution of $n$), so that the average size of a synchronized population $<n>$ and entropy $S$ within any interval of duration $d$ are given by,

$$\langle n \rangle = \int_0^\infty f(n) \cdot n \, dn = \frac{1}{p_{term.d}}, \qquad \text{Eq5}$$

and

$$S = -\int_0^\infty f(n) \cdot ln\{f(n)\} \, dn = 1 - ln(p_{term.d}), \qquad \text{Eq6}$$

where $f(n)$ is given by *Eq 3* with $p_{term.d}$ substituted for $p_{term}$. Note that even if $p_{term.d}$ varied beyond the level of stochastic fluctuations (*Eq 4*), *Eqs 5* and *6* would still hold if $p_{term.d}$ remained constant on the timescale of $d$.

Taking the natural log of *Eq 5*, we get, $ln(\langle n \rangle) = -ln(p_{term.d})$. Substituting this into *Eq 6* yields a linear relationship between entropy, $S$, and the log average size of the synchronized population, $ln(<n>)$,

$$S = ln(\langle n \rangle) + 1. \qquad \text{Eq7}$$

As we assume that the observed spectral power, $SP$, at each site is proportional to the size of the synchronized neural population $n$, we have

$$SP = k \cdot n, \qquad \text{Eq8}$$

where $k$ is the constant of proportionality. Taking the temporal average yields,

$$\langle SP \rangle = k \cdot \langle n \rangle. \qquad \text{Eq9}$$

Taking the natural log of *Eq 9*, $ln(\langle SP \rangle) = ln(\langle n \rangle) + ln(k)$, and solving for $ln(<n>)$, we get,

$$ln(\langle n \rangle) = ln(\langle SP \rangle) - ln(k). \qquad \text{Eq10}$$

Substituting *Eq 10* into *Eq 7* yields,

$$S = ln(\langle SP \rangle) + 1 - ln(k). \qquad \text{Eq11}$$

Note that any attenuation of $SP$ due to the use of scalp-recorded EEG to compute spectral power is absorbed in $k$ (Eq 8). The computation of entropy, $S$, requires binning of spectral-power values (see Materials and methods) to generate their probability distribution per $d$ (sec) interval, which tends to underestimate entropy. We accommodated this underestimation of the true entropy, $S$, by the observed entropy, $S_{obs}$, by introducing a scaling factor $a$ and an additive term $b$,

$$S_{obs} = aS + b, \qquad\qquad\qquad \text{Eq12}$$

where $0 \leq a \leq 1$; $a$ approaches 1 and $b$ approaches 0 with a larger number of data points and finer spectral-power bins per interval. Substituting Eq 11 into Eq 12 yields,

$$S_{obs} = a \cdot [ln(\langle SP \rangle) + 1 - ln(k)]. \qquad\qquad \text{Eq13}$$

The parameter $b$ has been absorbed in $k$ because linear fitting cannot distinguish between $b$ and $k$. As such, the observed value of $k$ would be difficult to interpret.

Phase-scrambled spectral-power dynamics (which we have confirmed to obey Eq 3; Fig 2) should obey Eq 13 for intervals of any duration $d$ (given that it includes sufficient data points to reliability evaluate the probability distribution of $SP$). To confirm this prediction, we divided each ~5 min EEG recording period into non-overlapping $d$ (sec) intervals and computed average spectral power $<SP>$ and entropy $S_{obs}$ for each interval. The use of a longer interval, providing a larger number of $SP$ values per interval, would make the relationship between $ln(<SP>)$ and $S_{obs}$ tighter by increasing the accuracy of estimating $<SP>$ and $S_{obs}$. However, it would reduce the variability in $<SP>$ and $S_{obs}$ across intervals (Eq 4) and also reduce the number of $ln(<SP>)$-$S_{obs}$ pairs to evaluate their relationship over time. We present our primary analyses with $d = 3$ sec; the choice of this particular duration will be justified below.

Each upper-left panel in Fig 3 shows, for the phase-scrambled controls, the 2D-density plot of $ln(<SP>)$-$S_{obs}$ pairs for all $d = 3$ sec intervals for all frequencies (i.e., wavelet center frequencies), sites, and participants for a specific condition. Density is color-coded as percentile so that confidence intervals can be inferred. As predicted by Eq 13, the relationship between $ln(<SP>)$ and $S_{obs}$ for the phase-scrambled controls was linear for all conditions, rest with the eyes closed, its replication, rest with the eyes open in dark, and the earlier and later viewing of a silent nature video.

We note that minor deviations from linearity occurred in the extreme ranges of spectral power for technical reasons. First, the binning of spectral-power values necessary to compute entropy per time interval (see Materials and methods) caused an underestimation of entropy, generating the slight upward curvature in the lowest spectral-power range especially for $ln[<SP>] < 0$ (see the lower-left portions of the left panels in Fig 3) due to the floor effect (entropy > 0). Second, the use of a fixed maximum spectral-power bin (necessary to compute entropy over the same range of spectral-power bins in all cases) prevented extremely high-power values (though up to at least 99.9[th] percentile of the values were retained; see Materials and methods) from contributing to the computation of entropy, causing an underestimation of entropy in the highest spectral-power range especially for $ln[<SP>] > 5.5$ (see the upper-right portions of the left panels in Fig 3). These extreme ranges of $ln[<SP>]$ were excluded from the subsequent analyses (also from the computation of the linear fits shown in Fig 3).

Notwithstanding these minor deviations for the extreme values of $ln(<SP>)$, the crucial observation is that the relationship between $ln(<SP>)$ and $S_{obs}$ were consistently linear for all conditions for the phase-scrambled controls, obeying Eq 13. Because Eq 13 derives from Eq 3 (describing a maximum-entropy distribution), the linear relationships defined by the phase-scrambled controls indicate the *line of maximum entropy*.

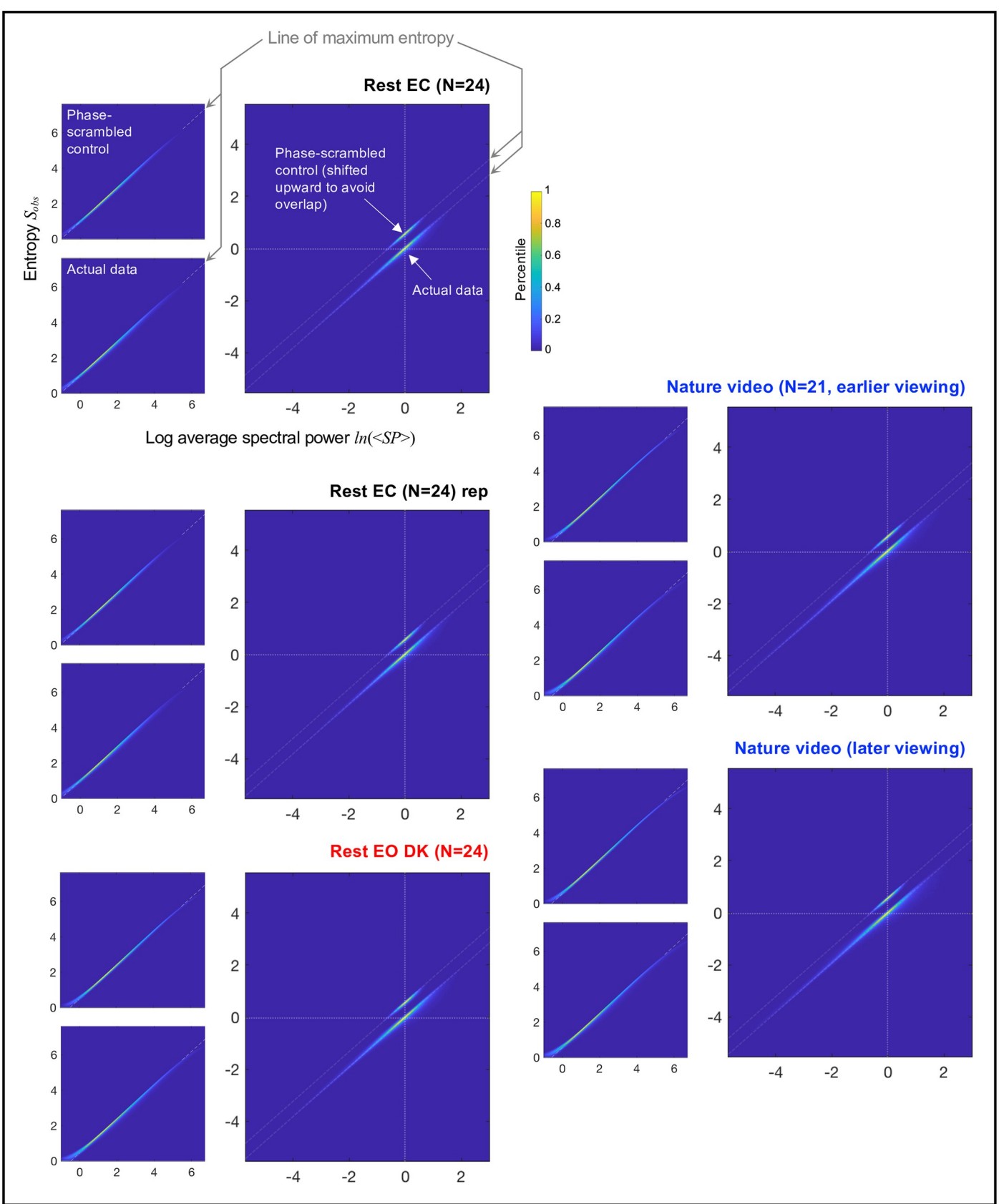

**Fig 3. Relationship between log average spectral power $ln(\langle SP \rangle)$ and entropy $S_{obs}$ for $d$ = 3 sec intervals.** Each set of three panels shows the data for a specific condition: ~5-min rest with the eyes closed (Rest EC), its replication (Rest EC rep), ~5-min rest with the eyes open in dark (Rest EO DK), and the earlier and later ~5-min viewing of a silent nature video (Nature video). For each condition, the $ln(\langle SP \rangle)$-$S_{obs}$ pair was computed for each non-overlapping $d$ = 3 sec interval per frequency per site per participant. **Upper-left panels.** 2D-density plots of all $ln(\langle SP \rangle)$-$S_{obs}$ pairs for the phase-scrambled controls. The linear fits (gray dashed oblique lines) indicate the line of maximum entropy indicative of spectral power fluctuations that maximize entropy for a given value of average spectral power (see text). **Lower-left panels.** 2D-density plots of all $ln(\langle SP \rangle)$-$S_{obs}$ pairs for the actual EEG data. Note that the distributions follow the line of maximum entropy (the gray dashed oblique lines) defined by the phase-scrambled controls. **Main panels.** Re-plotting of the 2D-density plots for both the phase-scrambled controls and the actual EEG data after aligning the phase-scrambled 2D-density plot for each frequency, site, and participant at its center at (0,0) and equivalently translating the corresponding actual-data density plots. The 2D-density plots for the phase-scrambled controls are shifted upward to avoid overlaps with those for the actual EEG data. The centering shows that the dynamic ranges of average spectral power (per $d$ = 3 sec interval) were substantially extended along the line of maximum entropy (the gray dashed oblique lines) for the actual EEG data relative to their phase-scrambled controls in all conditions. This pattern was observed for all representative frequencies (Fig 4) and all participants (S1 and S2 Figs). Thus, on the timescale of up to about 3 sec, spectral power appears to be controlled in such a way that the dynamic ranges are substantially extended (relative to stochastic dynamics) while tightly conforming to the line of maximum entropy. **All panels.** Density is color-coded as percentile so that confidence intervals can be inferred. The extreme ranges of spectral power, $ln(\langle SP \rangle) < 0$ and $ln(\langle SP \rangle) > 5.5$ were excluded from the computations of the line of maximum entropy and the centered 2D-density plots (the main panels) to avoid the binning-related distortions (see text).

Remarkably, the relationship between $ln(\langle SP \rangle)$ and $S_{obs}$ for the actual EEG data tightly clustered along the line of maximum entropy (the lower-left panels in Fig 3). Because average spectral power considerably varied depending on frequency, sites, and participants, the range of temporal variations in $ln(\langle SP \rangle)$ and $S_{obs}$ are obscured when $ln(\langle SP \rangle)$-$S_{obs}$ pairs from all frequencies, sites, and participants are plotted together. To focus on the temporal variation in $ln(\langle SP \rangle)$ and $S_{obs}$, we aligned the 2D-density plot for the phase-scrambled control for each frequency, site, and participant at its center at (0,0) and equivalently translated the density plots for the corresponding actual EEG data.

The centered relationships between $ln(\langle SP \rangle)$ and $S_{obs}$ are shown in the main panels in Fig 3. The 2D-density plots for the phase-scrambled controls are shifted upward to avoid overlaps with those for the actual EEG data, with the parallel gray dashed oblique lines indicating the line of maximum entropy. It is clear that the ranges of average spectral power $\langle SP \rangle$ (for $d$ = 3 sec intervals) were substantially extended in the actual EEG data relative to their phase-scrambled controls while consistently following the line of maximum entropy. This pattern appears to be universal, observed in all conditions (the main panels in Fig 3), all representative frequencies per condition (Fig 4), and all participants (S1 and S2 Figs). These results suggest that spectral-power dynamics maintain maximum entropy on the timescale of a few seconds (Eqs. 5–6) while generating large power variations (relative to phase-scrambled controls) by changing the probability of synchronization on slower timescales.

To examine how closely the spectral-power dynamics followed the line of maximum entropy we computed the probability distributions of entropy around the line of maximum entropy for the actual EEG data and their phase-scrambled controls. While the line of maximum entropy was virtually identical for all participants (e.g., S1 and S2 Figs), here we computed it separately for each participant to increase the accuracy in estimating entropy distributions around it. The bins to compute the entropy distributions were determined by the maximum (*max*), minimum (*min*), and inter-quartile range (*iqr*) of the entropy distributions for the relevant EEG data and their phase-scrambled controls (per participant), with the number of bins given by, $N_{bins} = ceil\left(\frac{max-min}{2 \cdot iqr \cdot N_{intervals}^{-1/3}}\right)$ (Freedman & Diaconis, 1981), where $N_{intervals}$ is the number of $d$ (sec) intervals for which $ln(\langle SP \rangle)$-$S_{obs}$ pairs were computed.

These probability distributions are plotted in Fig 5A for the five conditions for representative interval durations, $d$ = 1, 3, 10, 20, 40, and 90 sec. The negative and positive values on the $x$-axis indicate the negative and positive deviations from the line of maximum entropy normalized to the standard deviation of the distribution for the corresponding phase-scrambled controls, and the $y$-axis indicates probability density. The shaded areas represent the distributions for the phase-scrambled controls (symmetric about the line of maximum entropy regardless of

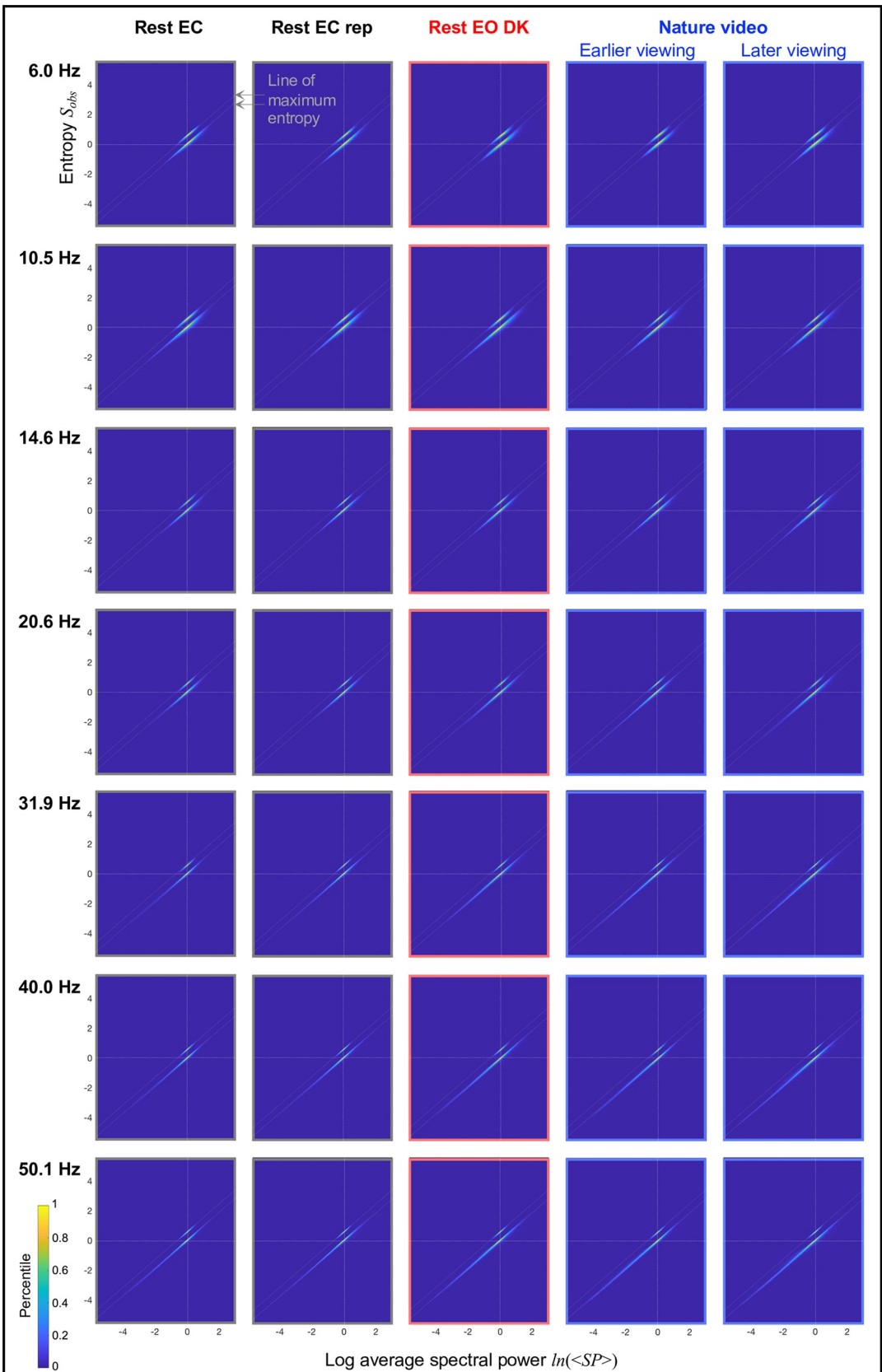

**Fig 4. The same as the main panels in Fig 3, but the centered relationships are shown separately for the representative frequencies (rows) and conditions (columns).** The phase-scrambled distributions are shifted upward to avoid overlaps with the actual-data distributions, and the gray dashed oblique lines indicate the line of maximum entropy. Note that the dynamic ranges of average spectral power were substantially extended along the line of maximum entropy for the actual EEG data relative to their phase-scrambled controls for all representative frequencies in all conditions.

*d* as expected), the solid curves represent the distributions for the actual EEG data, and the solid areas represent the regions where the probability density was higher for the actual data than for their phase-scrambled controls.

The actual and phase-scrambled distributions were virtually indistinguishable for $d = 1$ sec and $d = 3$ sec for all five conditions (the two left columns in Fig 5A), indicating that the actual data tightly followed the line of maximum entropy up to $d = 3$ sec. For longer interval durations, the actual distributions progressively extended in the lower-entropy direction, seen as the solid-colored negative tails increasing in the third through the last column in Fig 5A. We quantified these lower-entropy tails for the actual EEG data by computing the *proportions of lower-entropy intervals* (**PrLEI**) for the actual data relative to their phase-scrambled controls. Specifically, for each distribution we computed the actual minus phase-scrambled probability density wherever the density was higher for the actual data than for the phase-scrambled controls, and summed those differences (multiplied by the bin width to convert to proportions) separately on the negative and positive sides, then subtracted the sum on the positive side from the sum on the negative side. This algorithm essentially yielded the proportion of the actual-data distribution extending in the lower-entropy direction relative to the corresponding phase-scrambled distribution while compensating for any changes in distribution widths (approximately the area proportion of the solid-colored negative tails of the actual data in Fig 5A). For example, PrLEI = 0.2 would indicate that for a given interval duration *d*, the occurrences of lower-entropy intervals for the actual data were 20% more frequent than for their phase-scrambled control.

We computed PrLEI values per participant per condition and plotted them as a function of interval duration *d* in Fig 5B. The circular symbols connected with thick lines indicate the median PrLEI values with the thin dotted lines showing the values for the individual participants. While PrLEI became large for longer interval durations (note the *y*-axis is reversed), the median PrLEI values remained small ($< 5\%$) and condition independent up to about $d = 3$ sec. This indicates that up to the timescale of a few seconds, only up to about 5% of intervals of the actual EEG data (on average) more negatively deviated from the line of maximum entropy than their phase-scrambled controls. That is, on average, greater than 95% of spectral-power dynamics followed the line of maximum entropy on the timescale of a few seconds. Even at the level of individual participants, only a few (out of 52), yielded PrLEI values greater than 10% for $d = 3$ sec (the dotted lines in Fig 5B).

The PrLEI values (for $d = 3$ sec) were consistently low for all frequencies for all conditions (Fig 6A) and globally low at all sites for all conditions (Fig 6B). Nevertheless, the data potentially suggest an interesting spatial pattern. We *z*-transformed the PrLEI values across sites per participant to quantify the consistency of regional deviations in PrLEI from the spatial average as *t*-values (with $|t| > 3.95$ for Bonferroni-corrected 2-tailed significance at $\alpha = 0.05$) (Fig 6C). Cooler colors indicate regions with lower-than-average PrLEI values while warmer colors indicate regions with higher-than-average PrLEI values. In the eyes-open conditions, entropy was near maximal (very low PrLEI values) in the mid-central-posterior region (the dark blue regions highlighted with dotted circles in the lower three rows of Fig 6B and 6C). At the same time, consistent elevations in the PrlEI values (though still low with the means of less than 8.7% for all sites for all conditions) were observed in areas surrounding the mid-central-

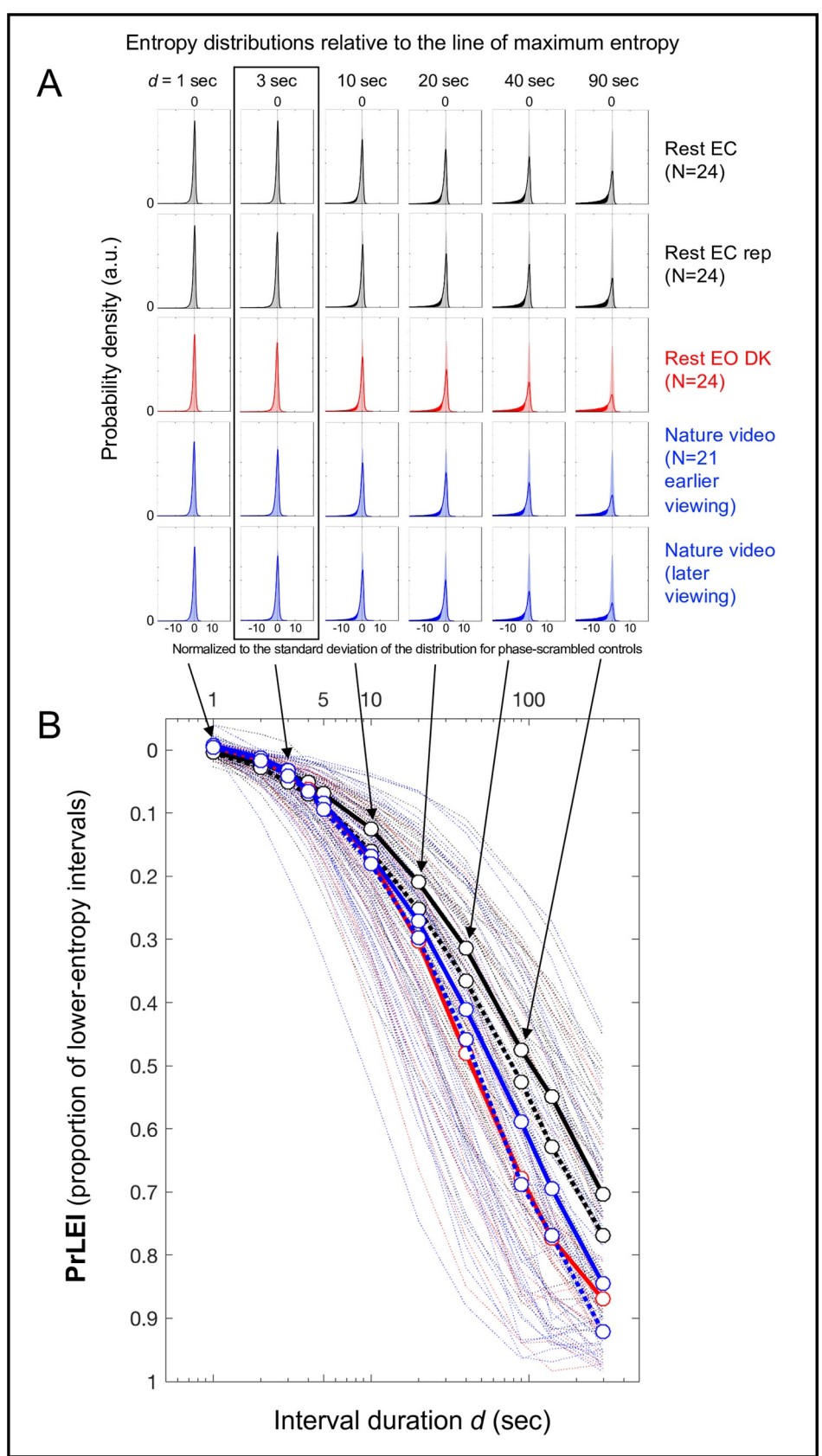

**Fig 5. Probability distributions of entropy $S_{obs}$ relative to the line of maximum entropy for the actual EEG data and their phase-scrambled controls as a function of interval duration $d$. A.** Probability distributions for the phase-scrambled controls (shaded areas) and actual EEG data (solid outlines), with the line of maximum entropy labeled as 0 on the $x$-axis and the negative tails of the actual-data distributions toward lower entropy shown as solid areas. The rows correspond to the five conditions and the columns correspond to the representative interval durations $d$ (sec). The x-axis of each probability distribution has been normalized to the standard deviation of the corresponding phase-scrambled-control distribution. Note that up to about $d$ = 3 sec (highlighted with a rectangle), the distributions for the actual EEG data and their phase-scrambled controls virtually overlap. **B.** Proportions of lower-entropy intervals (**PrLEI**) for the actual EEG data relative to their phase-scrambled controls (approximately the area proportion for the solid-colored negative tails shown in A) as a function of interval duration $d$ (sec). This measure indicates the proportions of $d$ (sec) intervals for which the actual EEG data yielded lower entropy than predicted by the line of maximum entropy. The circular symbols connected with thick lines indicate the median PrLEI values with the five conditions color-coded as in A (the black dotted lines indicating the replication of the rest-with-the-eyes-closed condition and the blue dotted lines indicating the later viewing of the nature-video condition) with the thin dotted lines showing the PrLEI values for the individual participants. Note that for the interval durations up to about $d$ = 3 sec the actual EEG data closely followed the line of maximum entropy with less than ~5% deviations (in median PrLEI values) across all conditions, suggesting that neural dynamics on the spatial-scale of EEG current sources generally maintain maximum entropy up to the timescale of a few seconds (see text).

posterior region (Fig 6C). In particular, in the eyes-closed conditions the PrLEI values were focally elevated in the right-lateral region (the upper two rows in Fig 6C).

## General discussion

The dynamics of EEG spectral power deviate from stochastic dynamics in a U-shaped manner, such that the occurrences of the lowest and highest ranges of power are elevated (Fig 2). We used a simple mathematical model of synchronization dynamics to investigate the mechanisms that generate these characteristic deviations.

We modeled synchronization dynamics as simple chain reactions, where synchronization sequentially spreads from neural-unit to neural-unit at each moment with the probability $p_{term}$ for the rapid spreading to terminate (Eq 1). Although one may question the physiological relevance of postulating synchronization to independently spread at each time moment, the model (Eq 1) is useful in the sense that it provides a simple computational mechanism that generates synchronization dynamics that maximize entropy for a given temporal average (Eq 3). Note that the inferences that we have drawn are valid irrespective of the physiological plausibility of the specific model because they are based on how EEG spectral-power dynamics obeyed the rule of maximum entropy (i.e., Eq 13 derived from Eq 3). If the simple chain-reaction model of synchronization dynamics (Eq 1) were physiologically relevant, the parameter $p_{term}$ could be interpreted as the probability of termination of the sequential spreading of synchronization. If not, $p_{term}$ would be related to the probability of synchronization in some other way with the quantity $1/p_{term}$ directly related to the temporal average of the size of synchronized neural population (Eq 5).

We assumed that EEG spectral power was proportional to the size of the synchronously oscillating neural population accessible at each site. For a constant $p_{term}$, the model predicted stochastic dynamics (Eq 3) with the temporal variation of spectral power exponentially distributed. The model further predicted that if $p_{term}$ remained constant the fluctuations of log average spectral power and entropy should be associated along the line of maximum entropy (Eq 13 derived from Eq 3) on any timescale. These predictions were confirmed for the phase-scrambled controls (Fig 5A). Eq 13 further predicted that even if $p_{term}$ substantially varied as a function of time, if it remained relatively constant up to some timescale $d$ (sec) the fluctuations of log average spectral power and entropy should still be associated along the line of maximum entropy up to that timescale (Eqs 5 and 6).

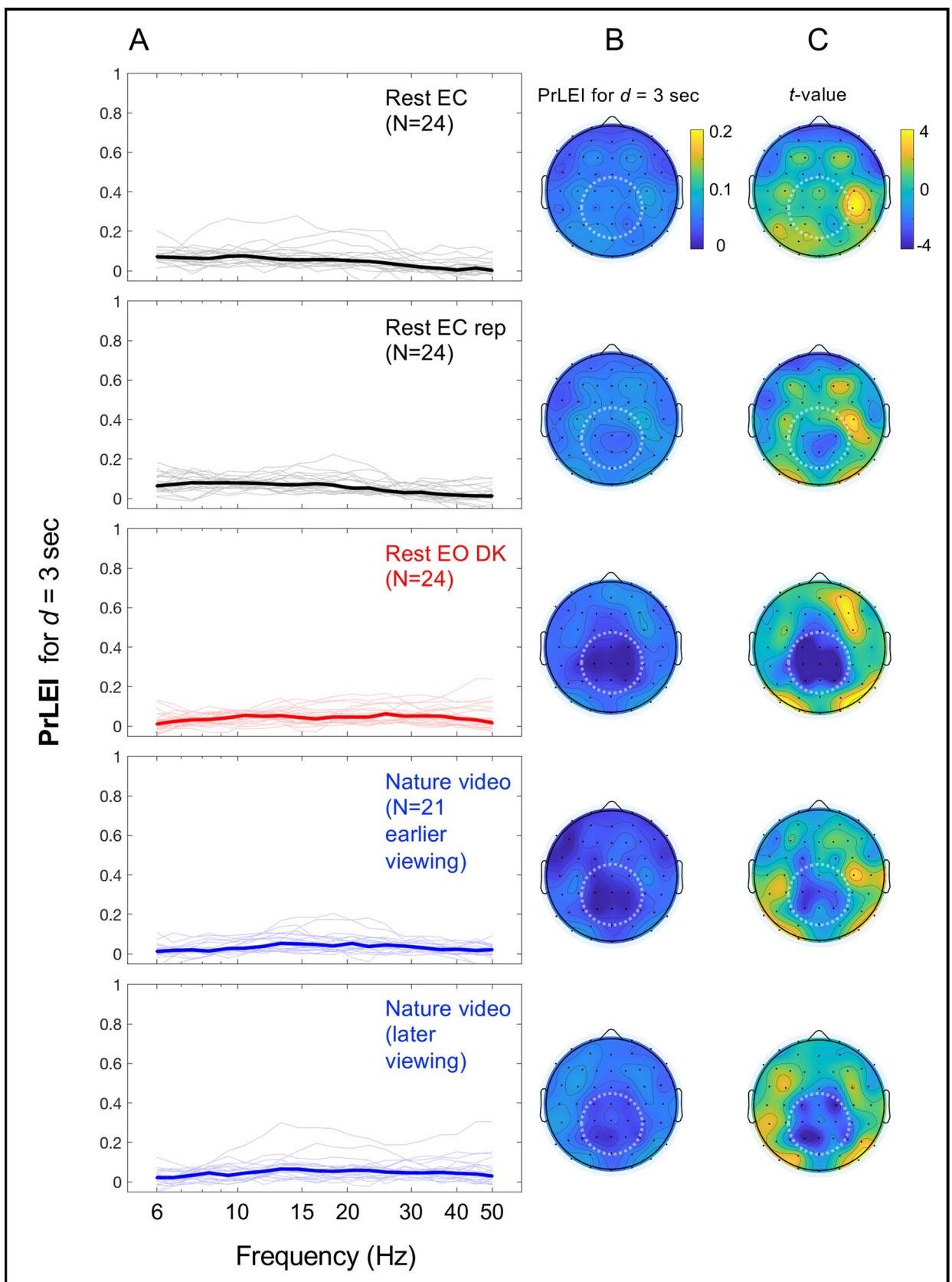

**Fig 6. Proportions of lower-entropy intervals (PrLEI) for $d$ = 3 sec for the actual EEG data relative to their phase-scrambled controls as a function of frequency and site.** PrLEI indicates the proportion of $d$ (sec) intervals for which the actual EEG data had lower entropy than predicted by the line of maximum entropy. **A.** PrLEI as a function of frequency. The thick lines indicate the median PrLEI values with the thin lines showing the values for the individual participants. The rows correspond to the five conditions. Note that the median PrLEI values were low regardless of frequency or condition. **B.** PrLEI as a function of site. The rows correspond to the five conditions as in A. The mean PrLEI values were globally low across all sites and conditions. The mid-central-posterior region (highlighted with dotted circles) yielded particularly low PrLEI values in the eyes-open conditions (the rest-with-the-eyes-open-in-dark and nature-video conditions) (the lower three rows). **C.** Same as B, but the data from each participant were z-transformed across sites to quantify the consistency of regional deviations of PrLEI values from the spatial average as $t$ values (with $|t|$>3.95 for Bonferroni-corrected 2-tailed significance at $\alpha$ = 0.05). Cooler colors indicate regions with lower-than-average PrLEI values while warmer colors indicate regions with higher-than-average PrLEI values. The $t$-values confirm that the PrLEI values were consistently low in the mid-central-posterior region in the eyes-open conditions (see B). Further, consistent elevations in the PrLEI values (though still low with the means of less than 8.7% for all sites and conditions) occurred in areas surrounding the mid-central-posterior region, particularly in the right-lateral region in the eyes-closed conditions (the upper two rows).

The results clearly showed that up to a few seconds ($d$ = 3 sec), the dynamics of EEG spectral power closely followed the line of maximum entropy (Fig 3) for all representative frequencies (Fig 4) and all participants (S1 and S2 Figs) whether they rested with their eyes closed, rested with their eyes open in a darkened room, or viewed a silent nature video. The EEG spectral-power dynamics were nearly as tightly distributed around the line of maximum entropy as were their phase-scrambled controls (see the $d$ = 3 sec column highlighted with the rectangle in Fig 5A). Any systematic deviations from the line of maximum entropy were small up to $d$ = 3 sec with the median PrLEI values remaining low for all frequencies and conditions, especially for the eyes-open conditions where the median values were consistently below ~5% (Fig 6A). These results suggest that the majority (~95%) of the low- and high-spectral-power states that deviated from stochastic dynamics (Fig 2) were generated by relatively slow mechanisms that generally maintain maximum entropy within the timescale of a few seconds while changing the probability of synchronization on slower timescales to substantially extend the dynamic range of spectral power along the line of maximum entropy (Figs 3 [the main panels], 4, and S1 and S2 Figs).

This conclusion may seem counterintuitive because precisely coordinated actions and mental operations require tight controls of oscillatory neural dynamics. One possibility is that the large-scale spatial synchronization of oscillatory activity may be controlled by a combination of the relatively slow, entropy-maximizing adjustments of synchronization probability and the temporally precise adjustments of phase such as phase-resetting generated by sensorimotor interactions. For example, the inter-regional and cross-frequency coordination of large-scale oscillatory activity may be generally controlled by slowly co-varying the probabilities of synchronization across the relevant regions and frequency bands while maintaining maximum entropy on the timescale of a few seconds for energy efficiency. At the same time, the precisely timed coordination of inter-regional and cross-frequency synchronization of oscillatory activity may utilize phase-resetting initiated by punctate sensorimotor signals such as those generated by multisensory environmental stimuli as well as goal-directed and routine sensorimotor events such as blinks, saccades, and active touch (e.g., [40–45]; see [46, 47] for reviews). Further, non-oscillatory neural activities obscured in frequency-decomposition approaches may also play a substantial role in precisely timed neural coordination.

While the deviations from the line of maximum entropy were globally low on the timescale of a few seconds at all sites for all conditions (Fig 6B), we observed some notable spatial patterns. In particular, in the eyes-open conditions entropy was near maximal in the mid-central-posterior region (highlighted with dotted circles in the lower three rows in Fig 6B and 6C). This potentially suggests that, especially in the presence of substantial sensory input (the eyes-open conditions here), the mid-central-posterior region plays the role of an "entropy dump" to facilitate the temporally precise control of spectral-power dynamics in the surrounding

regions. In the eyes-closed conditions, we observed focal PrLEI elevations in the right-lateral region (the upper two rows in Fig 6C), potentially suggesting that this cortical region plays a role in controlling synchronization dynamics for constructing spontaneous imagery and thoughts (which were encouraged in the rest-with-the-eyes-closed condition). While these interpretations are highly speculative, tracking deviations of spectral-power dynamics from the line of maximum entropy may provide a useful method for tracking the spatiotemporal occurrences of temporally precise controls of large-scale spectral dynamics.

While the current analyses used a time-frequency decomposition approach that extracted sinusoidal components from EEG, neuronal oscillations are not necessarily sinusoidal (though oscillatory activities of large neural populations, as reflected in EEG, tend to approximate sinusoidal waveforms due to spatial averaging [48]). Non-sinusoidal waveforms are detected by time-frequency decomposition methods as the sinusoidal components with the approximately matching frequencies as well as additional sinusoidal components at various harmonic frequencies that may spuriously mimic phase-to-amplitude coupling [e.g., 49, 50]. We observed entropy-maximizing behaviors on the timescale of a few seconds for sinusoidal components of a broad range of frequencies. Thus, even if some of the observed sinusoidal components reflected the fundamental and harmonic components of non-sinusoidal oscillations, we could still infer that the underlying non-sinusoidal oscillations exhibit entropy-maximizing behaviors on the timescale of a few seconds in the sense that their sinusoidal components do.

In summary, we used a simple mathematical model of synchronization to investigate the mechanisms that make EEG spectral-power dynamics deviate from stochastic dynamics in a characteristic U-shaped manner (Fig 2). The results have clearly shown that the majority (~95%) of this control is universally (across frequencies, sites, and behavioral conditions) accomplished by slowly changing the probability of synchronization while maintaining maximum entropy on the timescale of a few seconds. The results may further suggest that the mid-central-posterior region potentially serves as an entropy dump to facilitate the generation of precisely controlled spectral-power dynamics in the surrounding regions.

## Supporting information

**S1 Fig. Individual participants' data for the main panels in Fig 3 for participants p1-p28 who participated in the rest-with-the-eyes-closed condition (Rest EC), the nature-video condition (Nature video), or both.** All participants who participated in the nature-video condition provided data for both earlier and later viewings. The dynamic ranges of average spectral power (for $d$ = 3 sec intervals) were moderately to substantially extended along the line of maximum entropy (the gray dashed oblique lines) for the actual EEG data relative to their phase-scrambled controls for all participants for all conditions. Note that the degree of extension of spectral-power dynamic range does not appear to be a trait-like property as it substantially varied across conditions (even between the two instances of the nature-video condition) for some participants.
(TIF)

**S2 Fig. Individual participants' data for the main panels in Fig 3 for participants p29-p52 who participated in the replication of the rest-with-the-eyes-closed condition (Rest EC rep) and the rest-with-the-eyes-open-in-dark condition (Rest EO DK).** The dynamic ranges of average spectral power (for $d$ = 3 sec intervals) were moderately to substantially extended along the line of maximum entropy (the gray dashed oblique lines) for the actual EEG data relative to their phase-scrambled controls for all participants for all conditions. Note that the degree of extension of spectral-power dynamic range does not appear to be a trait-like

property as it substantially differed between the two similar conditions for some participants. (TIF)

## Author Contributions

**Conceptualization:** Satoru Suzuki.

**Data curation:** Melisa Menceloglu.

**Formal analysis:** Satoru Suzuki.

**Funding acquisition:** Melisa Menceloglu.

**Investigation:** Melisa Menceloglu, Marcia Grabowecky, Satoru Suzuki.

**Methodology:** Melisa Menceloglu, Satoru Suzuki.

**Project administration:** Marcia Grabowecky, Satoru Suzuki.

**Resources:** Melisa Menceloglu, Marcia Grabowecky, Satoru Suzuki.

**Software:** Satoru Suzuki.

**Supervision:** Marcia Grabowecky, Satoru Suzuki.

**Validation:** Melisa Menceloglu, Satoru Suzuki.

**Visualization:** Satoru Suzuki.

**Writing – original draft:** Satoru Suzuki.

**Writing – review & editing:** Melisa Menceloglu, Marcia Grabowecky, Satoru Suzuki.

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
