## [Decision Letter · Decision Letter 0]

21 Jan 2021

PONE-D-20-26112

Probabilistic, entropy-maximizing control of large-scale neural synchronization

PLOS ONE

Dear Dr. Suzuki,

Thank you for submitting your manuscript to PLOS ONE. After careful consideration, we feel that it has merit but does not fully meet PLOS ONE’s publication criteria as it currently stands. Therefore, we invite you to submit a revised version of the manuscript that addresses the points raised during the review process.

In particular one of the reviewers have concerns about the organization of your MS as well as the justification of some of the theoretical developments. You must address these points.

We look forward to receiving your revised manuscript.

Kind regards,

Pedro Antonio Valdes-Sosa, Ph.D., M.D.

Academic Editor

PLOS ONE

Journal Requirements:

2. Please provide additional details regarding participant consent.

In the ethics statement in the Methods and online submission information, please ensure that you have specified what type you obtained (for instance, written or verbal, and if verbal, how it was documented and witnessed).

If your study included minors, state whether you obtained consent from parents or guardians.

If the need for consent was waived by the ethics committee, please include this information.

Reviewers' comments:

Reviewer's Responses to Questions

**Comments to the Author**

1. Is the manuscript technically sound, and do the data support the conclusions?

Reviewer #1: Yes

Reviewer #2: No

2. Has the statistical analysis been performed appropriately and rigorously? 

Reviewer #1: Yes

Reviewer #2: I Don't Know

3. Have the authors made all data underlying the findings in their manuscript fully available?

Reviewer #1: Yes

Reviewer #2: No

4. Is the manuscript presented in an intelligible fashion and written in standard English?

Reviewer #1: Yes

Reviewer #2: No

5. Review Comments to the Author

Reviewer #1: In this paper, the authors used a simple chain model of spreading synchronization to account for deviation from purely stochastic dynamics in spontaneous EEG under various experimental conditions. The link between the model and the EEG data was constructed by the assumption that the size of a synchronized region is proportional to the observed spectral EEG power. This is a reasonable assumption.

They showed that (1) spectral power distribution of real EEG deviates from its stochastic (phase scrambled) counterpart in a u shaped manner, elevating high and low powers, (2) both stochastic and real EEG obey the maximum entropy relationship predicted by the model for a constant probability of spreading termination, (3) for longer time windows above 3 seconds, however, there was a deviation, indicating that the deviation of real EEG in the low and high power ranges are associated with relatively slow mechanisms, and (4) that these deviations from stochastic behavior consistently featured particular spatial patterns across brain regions (only rough, because surface Laplacian was used). The authors offer some (fairly speculative) explanation for the meaning of these findings.

This study is carefully conducted and yielded interesting and (in my view) highly relevant results that could also help to build and understand more biologically realistic models of neural dynamics. In addition, the presented methodology may be very useful for further research. As far as I can judge, all theory is consistent and well founded. The paper is very well written and instructive. I enjoyed reading it.

As a conclusion, I am inclined to recommend unconditional acceptance (which I very rarely do). Nonetheless, I have a few suggestions for the authors:

1) In the introduction, especially in the first paragraph, it sounds as if brain dynamics is exhaustively described by interacting (harmonic) oscillations. What about transient dynamics and oscillations that are not at all sinusoidal? Detailed neural and neural population models predict this. Does the Fourier decomposition not generate a biased representation here? Please comment.

2) Page 13, last para: typo “2D-density”

3) Fig. 5A: please indicate the scaling of the x axes.

Reviewer #2: This manuscript is about the mechanisms underlying the large scale synchronization of the brain activity. This is an important and interesting issue that deserves a careful analysis with well defined and documented hypotheses. The reviewer does not reject the overall approach proposed by the authors but rather invite them to revise in depth the rationale of the work. There is a huge confusion between the a priori model, the 'thermodynamic' point of view (entropy, energy, adiabatic transform) and the results. There is too much vague statements and the mathematics, although very simple in terms of the developments, deserve some justifications.

I details, the comments are the following (they aimed at helping the resubmission of this work)

- Introduction: the authors should explain how the 'spectral power dynamics' (the original one and the phase-resampled ones) can help to explain synchronization dynamics. Of course, phase scrambling is a way to destroy coherence between harmonic component (distortion) but a simulation would be convincing.

The reviewer did not understand how 'the phase-scrambled spectral-power dynamics are characterized by exponential power distributions': distribution over what? The dynamics raises the issue of stationarity of the recordings. How much such a distribution is reliable?

The authors introduce the 'probability of synchronization' and the principle of 'maxEnt' together with the 'energy efficiency'. All those notions should be carefully defined in the present context.

Finally, this section should give the hypotheses underlying the objectives of the work.

- Methods:

To be able to reproduce the results, some of the methodological steps deserve more justification.

1) What was the criteria to select the ICA component(s) to be removed?

2) Please specify the parameters of the Current Source Density filtering of the EEG. This is an interesting and important pre-processing. What is the impact on the results? The authors should stress that without solving explicitly an inverse problem to map the EEG onto the cortical substrate, they tend to describe the underlying source activities. This is a good idea that should be more emphasized.

3) The EEG temporal derivative is nothing but a high-pass filter. The justification with the 'trigonometric properties' is a bit naive and the (scaling-) exponent beta seems to be neglected although it may be different from 1 and even changing with time. Why does the authors do not use the next coming wavelet framework to 'liter out' this scaling property?

4) Spectral power as a function of time. This is a difficult problem that raises, as previously mentioned, the issue of stationnarity. The time-frequency indeed offers a framework to handle this problem but the method is not described neither discussed in terms of the choice of the wavelet itself (a DOG wavelet would have been more appropriate). The author should better describe the method here.

5) The entropy introduced in 3.4 is not justified and the material given in the first pages of the results should be placed in the method. Jumping in this part (Results), the authors should stat with the Poisson distribution (Eq.3) to describe the neural population participating to a synchronized process (so giving a significant amplitude somewhere in the power spectrum). It was not clear for the reviewer why 'the exponential form of [the Poisson density] indicates that the model yields maximum entropy for fluctuations n, for a given temporal average <n>'? The figure2 deserves more explanation about the origin of the horizontal index (n?) and what we understand with 'residual deviation from...' (something to do with the Kullback-L divergence metric?). The exact definition of the defendant variable should be given.

6) It sounds that the objective of this methodological development is Eq.13. The relationship between the entropy introduced in 3.4 (positive sign because of the discrete nature of the p's ?) and the entropy introduced in Eq.6 is not clear.

7) The duration 'd' has something to do with an assumption of stationarity. Shouldd be commented.

8) The maximum entropy principle is not clear in the development

The Results are difficult to appreciate and the Discussion is rather speculative (and it sounds strange to see an hypothesis expressed in this section). The parameter p_term should be better interpreted (and justified). The interpretation as a probability is far from its introduction in (3)

If the main result is about the 'EEG dynamics that closely follow the line of Max Entropy for all representative frequencies', the manuscript is not convincing, neither the figures (3) an (4).

The reviewer invite the authors to revisit this work that may have a potential impact with a more rigorous presentation and argumentation. The reviewer encourage such work with clear statements about synchrony, maximum entropy principle and stationarity in EEG recordings.</n>

6. PLOS authors have the option to publish the peer review history of their article (what does this mean?). If published, this will include your full peer review and any attached files.

Reviewer #1: **Yes: **Thomas R. Knösche

Reviewer #2: No

---

## [Author Response · Author response to Decision Letter 0]

11 Feb 2021

Our point-by-point response to each of the reviewers' comments is attached as a PDF file.

---

## [Decision Letter · Decision Letter 1]

16 Mar 2021

Probabilistic, entropy-maximizing control of large-scale neural synchronization

PONE-D-20-26112R1

Dear Dr. Suzuki,

We’re pleased to inform you that your manuscript has been judged scientifically suitable for publication and will be formally accepted for publication once it meets all outstanding technical requirements.

Kind regards,

Pedro Antonio Valdes-Sosa, Ph.D., M.D.

Academic Editor

PLOS ONE

Additional Editor Comments (optional):

Reviewers' comments:

Reviewer's Responses to Questions

**Comments to the Author**

1. If the authors have adequately addressed your comments raised in a previous round of review and you feel that this manuscript is now acceptable for publication, you may indicate that here to bypass the “Comments to the Author” section, enter your conflict of interest statement in the “Confidential to Editor” section, and submit your "Accept" recommendation.

Reviewer #1: All comments have been addressed

2. Is the manuscript technically sound, and do the data support the conclusions?

Reviewer #1: (No Response)

3. Has the statistical analysis been performed appropriately and rigorously? 

Reviewer #1: (No Response)

4. Have the authors made all data underlying the findings in their manuscript fully available?

Reviewer #1: (No Response)

5. Is the manuscript presented in an intelligible fashion and written in standard English?

Reviewer #1: (No Response)

6. Review Comments to the Author

Reviewer #1: (No Response)

7. PLOS authors have the option to publish the peer review history of their article (what does this mean?). If published, this will include your full peer review and any attached files.

Reviewer #1: No

---

## [Editor Report · Acceptance letter]

23 Apr 2021

PONE-D-20-26112R1 

Probabilistic, entropy-maximizing control of large-scale neural synchronization 

Dear Dr. Suzuki:

I'm pleased to inform you that your manuscript has been deemed suitable for publication in PLOS ONE. Congratulations! Your manuscript is now with our production department. 

Kind regards, 

on behalf of

Professor Pedro Antonio Valdes-Sosa 

Academic Editor

PLOS ONE